# High Prevalence of Respiratory Symptoms among Particleboard Workers in Ethiopia: A Cross-Sectional Study

**DOI:** 10.3390/ijerph16122158

**Published:** 2019-06-18

**Authors:** Akeza Awealom Asgedom, Magne Bråtveit, Bente Elisabeth Moen

**Affiliations:** 1Center for International Health, Department of Global Public Health and Primary Care, Faculty of Medicine, University of Bergen, 5009 Bergen, Norway; bente.moen@uib.no; 2Ethiopian Institute of Water Resources, Addis Ababa University, Addis Ababa, P.O. Box 150461, Ethiopia; 3Department of Global Public Health and Primary Care, Faculty of Medicine, University of Bergen, 5020 Bergen, Norway; magne.bratveit@uib.no

**Keywords:** Ethiopia, lung function, particleboard factory workers, respiratory symptoms

## Abstract

Work in the wood industry might be associated with respiratory health problems. The production of particleboard used for furniture making and construction is increasing in many countries, and cause dust, endotoxin and formaldehyde exposure of the workers. The aim of the study was to assess the prevalence of respiratory symptoms and to measure lung function among Ethiopian particleboard workers using Eucalyptus trees as the raw material. In total 147 workers, 74 from particleboard production and 73 controls, participated in the study. Mean wood dust in the particleboard factories was measured to be above recommended limit values. Particleboard workers had a mean age of 28 years and the controls were 25 years. They had been working for 4 and 2 years, respectively. Lung function test was done using spirometry following American Thoracic Society (ATS) recommendations. Respiratory symptoms were collected using a standard questionnaire of ATS. Particleboard workers had higher prevalence of wheezing, cough, cough with sputum production, phlegm, and shortness of breath compared to controls. Lung function status was similar in the two groups. The symptoms might be related to the work in the factories. Longitudinal studies are recommended to explore the chronic impact of work in particleboard factories on respiratory health.

## 1. Introduction

Wood dust is a complex substance and one of the hazards generated from processing of various wood types for a wide range of applications [1,2]. Workers exposed to wood dust may develop different respiratory health problems [3,4] including reduced lung function [5,6,7]. An endotoxin component in the cell walls of gram-negative bacteria [8,9] might be present as a part of organic dust and may induce inflammatory responses in the airways [8,10,11]. In addition, formaldehyde that is added to the urea resin for gluing of wood products is associated with respiratory health problems [12,13] and decrements in lung function [14]. A range of biologically active compounds like quinones, terpenes, stilbenes, phenols, tannins, and flavonoids might also be released during wood processing [2].

In Ethiopia the manufacturing sector, comprising wood, metal, food, textile, leather and construction industries, accounts for 6.9% of the national work force [15]. Eucalyptus, an evergreen hardwood tree, is the main raw material for production of particleboard in Ethiopia [16] and is used for furniture like office tables and shelves, for interior wall partitioning [17,18] and construction [17,19]. The Eucalyptus tree is cheap, locally available, has rapid growth, and is adaptable to a range of climates and soil types. This makes it a promising source of inputs as the native wood species are diminishing due to deforestation. Despite the increasing production of furniture in Ethiopia [16] little is known about safety measures and occupational health in these workplaces [20]. In previous studies from the wood industry including particleboard production, the workers have been exposed to other types of trees [6,21]. More knowledge on the respiratory health of the particleboard workers exposed to dust from the Eucalyptus tree is needed to evaluate the need of occupational preventive measures in Ethiopia and other developing countries.

The international literature about the prevalence of respiratory health symptoms among wood workers varies greatly. For example, the prevalence of respiratory symptoms reported in Thailand and Iran varies from 15.5% to 41% [3,7]. Decrement in lung function among wood workers is reported in studies done in Thailand, Pakistan, Iran and Sweden [3,6,7,21,22], while other studies done in Poland and Denmark do not show any effect on lung function [23,24]. Thus, the international literature is not conclusive regarding the respiratory health for wood workers, and a study among particleboard workers using Eucalyptus trees as raw materials in Ethiopia is needed. 

The aim of this study was to assess the prevalence of respiratory symptoms and to measure lung function among particleboard factory workers of Ethiopia and compare the findings with a control group from water bottling factories with low exposure to dust. The findings might help to fill the research gaps on respiratory health among particleboard workers which can be applied to prevent respiratory disease at these workplaces.

## 2. Materials and Methods

### 2.1. Study Design and Period

A cross-sectional study was performed to assess respiratory symptoms and to measure lung function among workers from two of the largest particleboard factories in Ethiopia which use Eucalyptus trees as raw material. One of the factories was established in 2005, is situated in northern Ethiopia and has 663 workers. The other factory was established in 2002 and is located in southern Ethiopia and has 249 workers. The particleboard factories are found in urban areas and selected both from North and South. A control group was established of workers employed in a water bottling factory, with a total of 339 workers from northern Ethiopia. The controls were selected from a factory considered to have low dust concentration in the work environment. The study period of this paper was from May 2017 to August 2017.

### 2.2. Exposure Measurements 

Personal inhalable dust was sampled in the breathing zone of the workers using a conductive plastic inhalable conical sampler (CIS, JS Holdings, Stevenage, UK) [25] mounted with a 37 mm glass-fiber (GFA) filter (Whatman International Ltd, Maidstone, UK) using an air flow of 3.5 L/min Side Kick Casella (SKC) pump for 2 to 4 hours sampling duration per shift. In total, 76 workers in particleboard production were selected for repeated sampling of inhalable dust (*n* = 152). From the control group, i.e., the water bottling factory, 8 repeated samples were taken (*n*= 16). Inhalable dust was analyzed using gravimetric method in a room with controlled climatic conditions (22 °C, 45% relative humidity; desiccation ≥24 h) with an analytical balance with 0.1 µg readability (Mettler-Toledo Ltd, Greifensee, Switzerland) and the concentration was estimated in mg/m^3^. Endotoxin was analyzed using the Kinetic Amoebocyte Lysate test (Kinetic-QCL endotoxin kit, BioWhittaker, Walkersville, MA, USA) and the concentration was estimated in EU/m^3^.

The geometric mean, arithmetic mean and range of inhalable dust for particleboard workers was 4.66 mg/m^3^, 9.17 mg/m^3^, and 0.47–184 mg/m^3^, respectively. For the control group the figures were 0.21 mg/m^3^, 0.24 mg/m^3^, and 0.02–0.4 mg/m^3^, respectively. 

The geometric mean, arithmetic mean and range of endotoxin for particleboard workers was 62.2 EU/m^3^, 245.6 EU/m^3^, and 0.9–9202.2 EU/m^3^, respectively; while for the control group it was 0.66 mg/m^3^, 0.75 EU/m^3^, and 0.3–2.3 EU/m^3^, respectively. The concentration of formaldehyde measured with Dräger-Tubes by color tubes in the particleboard factories using “worst-case” sampling strategy ranged from <0.2 ppm to 5 ppm.

### 2.3. Study Population and Sample Size

The required number of participants for this study was calculated using OpenEpi software (http://www.openepi.com/SampleSize/SSMean.htm) sample mean difference by taking into consideration forced expiratory volume in one second (FEV_1_) as main output of interest with 95% power, 95% confidence interval and 5% level of significance. The FEV_1_ for exposed (3.77 L/s, SD = 0.99) and control group (4.29 L/s, SD = 0.86) was taken from a previously study done among particleboard workers in Ethiopia [26]. The estimated number of participants needed was 166 workers (83 from exposed and 83 from control groups).

### 2.4. Data Collection

To plan the study, the factories and their leadership were visited. After obtaining permission to perform the study, we asked the management to provide a list of workers in each work shift during the first phase of data collection as stated in a previously published paper [27] aimed to assess workers knowledge, attitude and practice regarding chemical hazards and personal protective equipment. Before the actual data collection, randomly selected participants were informed about the objective of the study, its relevance, and how to perform the interview and lung function measurements and asked for written consent to participate in the study.

#### 2.4.1. Respiratory Symptom Assessment

The interview of respiratory symptom assessment was done in a quiet and private place by six trained bachelor environmental health professionals.

Information on respiratory symptoms was collected using a validated standard questionnaire from the American Thoracic Society (ATS) [28]. The information collected were data on sex (M/F), age (years), education (highest grade completed), uses biomass fuel as sources of energy at home(Y/N), availability of separate kitchen (Y/N), number of service years in the present industry, occupational history in dusty working environment, and smoking (Y/N). The workers were also asked about their history of past respiratory illness (Y/N). If they had experienced any diseases, they were asked to tell what type it had been.

Questions about respiratory symptoms were asked like this; whether the workers in the last 12 months have/had: cough (Y/N), cough with sputum production (Y/N), phlegm (Y/N), episode of cough and phlegm (Y/N), wheezing (Y/N), shortness of breath (Y/N). The interviewed participants were also observed if they were using personal protective equipment (PPE) mainly face mask during the study.

The interview was based on questions prepared in English and translated to Amharic by a translator, and then translated back from Amharic to English by another translator, to check the consistency. Pretesting of the questionnaire was done on 5% of the sample population in one of the factories before the main study. The data collection tool was modified to suite the Ethiopian context. Information such as marital status and race were excluded from the questionnaire. Additional information about the use of biomass fuel as source of energy for cooking, availability of separate kitchen at home were added to the data collection tool. 

#### 2.4.2. Lung Function Test

Prior to performing the lung function measurements, the participants ID, age, sex, standing height (m) and weight (kg) were measured as recommended by the American Thoracic Society [29], and combined with the interview described in 2.4.1. Lung function test was done in sitting position using Spirometry (Minispir light with Winspiro software, Medical International Research (MIR), Rome, Italy) connected to a Laptop following American Thoracic Society guidelines [29]. The spirometry measurements were performed prior to the morning shift that started at 6:00 a.m. The lung function measurements were done by the trained researcher until the trial generated three acceptable maneuvers. From the three records of lung function test, the best trial was kept and used for the data analysis. The lung function parameters considered were FVC, FEV_1_, FEV_1_/FVC ratio and FEF_25–75%_. The FEV_1_/FVC ratio < 70% was the cutoff point for air flow limitations as stated by Global Initiative for Chronic Obstructive Lung Disease [30].

### 2.5. Data Management and Analysis

Collected data were checked for completeness and missing values at the end of each day of data collection. Data was exported from EpiData version 3.1 (EpiData Association, Odense, Denmark) to the statistical package SPSS, version 25 (International Business Machines Corporation (IBM), Armonk, NY, USA) for analysis. Lung function parameters were normally distributed. Descriptive statistics were used to summarize demographic and anthropometric data. Pearson chi-square or Fisher’s exact tests (if the expected value was less than 5) were used to test for difference in categorical variables, while an Independent *t*-test was used to compare means of continuous variables between exposed and controls. Poisson regression analysis was used to determine the prevalence ratio of cough between particleboard workers and water bottling workers while adjusting for educational status, previous work in dusty environment, age and availability of separate kitchen. Prevalence ratio (PR) was chosen instead of prevalence odds ratio (POR) due to the high respiratory symptom prevalence in this study [31].

Multiple linear regression was applied to analyze differences in lung function between the particleboard workers and the controls while adjusting for age, height, previous respiratory illness, availability of separate kitchen and use of biomass fuel as source of energy.

### 2.6. Ethical Approval

Ethical clearance for the study was obtained from the Regional Committee for Medical and Health Research Ethics, Western Norway on June 2, 2016 with IRB ref: IRB00006245 and from the Ethiopian Ministry of Science and Technology on October 7, 2016 with Ref. No. 3.10/148/2016. Study participants were informed about the purpose of the study, confidentiality of their information, duration of the interview, lung function measurement procedure and the possibility to withdraw from the study at any time they wanted. Written consent both from each of the study participants and consent from factory management was assured before data collection. 

The questionnaire and spirometry results were stored with only ID numbers of the participants, not their names. The data were locked in a safe place accessible only to the researcher to keep every person’s information confidential.

## 3. Results

### 3.1. Characteristics of the Study Population

From 166 people who were invited, 157 workers (94.5%) (83 particleboard workers and 74 water bottling workers) participated in the study. The remaining 5.5% did not want to participate in the study. Among the 157 who participated, one person from the particleboard workers could not properly perform the lung function measurement and we therefore discarded his lung function data. The majority, 147 (93.6%), of the participants in the study were males and used in the final analysis. Due to their low number, the females (5.7%)—eight participants from exposed and one participant from the water bottling factory—were excluded from the further analyses since gender differences are recognized for respiratory symptoms as well as for lung function [32,33,34]. The exposed group was older than the controls (28 vs. 25 years; *p* = 0.006) and had more service years than the controls (4 vs. 2 years; *p* < 0.001). The exposed groups were also more educated than controls (Table 1) and had higher body weight (63 vs. 56 kg; *p* < 0.001). The exposed and the control groups had the same average height (1.70 m). All of the study participants were neither smokers nor using proper personal protective devices such as face masks that can protect them from dust and other chemical exposures.

The majority (64%) of the exposed group had separate kitchens and only 30% used biomass fuel for cooking.

Some respondents in the exposed group had a history of previous illness such as bronchitis (*n* = 12), asthma (7), pneumonia (3) and tuberculosis (3), but such illnesses were not reported in the control group. In total, 18 (24%) participants had previous disease, of which 4 had more than one diagnosis.

### 3.2. Respiratory Symptoms

The prevalence of all recorded respiratory symptoms was significantly higher among the exposed (range 24–45%) than the controls (2.7–15%) (Table 2). A separate analysis was performed, excluding the participants from the particleboard factories who had previous respiratory diseases (*n* = 18). The results were still the same, except for cough which did not show any significant difference between the groups when these 18 persons were excluded (result not shown).

The prevalence ratio of cough among the exposed group was 1.56 (95% CI; 0.61, 3.95) compared to the controls when adjusted for education status, previous work in dusty environment, past history of respiratory illness, age, use of biomass fuel for cooking and availability of separate kitchen.

### 3.3. Lung Function

Lung function (FEV_1_/FVC) between the exposed group and the controls was significantly different (*p* = 0.004) when no adjustments were made (Table 3). 

However, in multiple regression models the difference in lung function between exposed and control groups was not significant when adjusting for age, height, previous respiratory illness, availability of separate kitchen and use of biomass fuel as source of energy (Table 3). All participants had FEV_1_/FVC values > 70%, indicating that none of the workers had airflow limitation. The same result was obtained when 18 participants with previous respiratory diseases were excluded from the analysis, both for the crude comparison of the groups and the regression analysis (result not shown).

## 4. Discussion

Workers in the particleboard manufacturing factories in Ethiopia had significantly higher prevalence of respiratory symptoms compared to water bottling workers (controls). The lung function values were not significantly different between the two groups when adjusting for age, height, previous respiratory illness, availability of separate kitchen and use of biomass fuel as source of energy.

The present study showed a higher prevalence of cough among particleboard workers than the controls. This finding agrees with studies done among wood workers in Tanzania, Macedonia, Iran and Sweden [3,4,21,35]. The increased prevalence of phlegm and wheezing among particleboard workers in our study was also in compliance with the findings among parquet manufacturing workers in Macedonia and sawmill workers in Iran [3,4]. Furniture manufacturing workers in Thailand had an increased risk of wheezing and breathlessness compared to office workers [7,12] which is also consistent with our finding.

A high prevalence of past history of respiratory illness (24%) was reported among particleboard workers but not in controls. However, the causes of past respiratory illnesses were not investigated. It may emanate from wood working activities or other dusty environments as some of the particleboard workers had worked in other dusty environment than the controls but can also be due to other unknown reasons.

Lung function was not significantly different between the particleboard workers and the control group. Our finding agrees with findings in Iran and Pakistan [3,22] which showed an insignificant difference in lung function between exposed and control groups. In our finding, all workers had a FEV_1_/FVC > 70%. This is similar with findings in Iran, Macedonia, Poland and Pakistan which showed that the mean FEV_1_/FVC ratio was higher than 70% among the study participants [3,4,5,22,23]. However, the result of our study is in contrast with studies done among Danish furniture workers which shows a reduced lung function [24]. According to the recommendation of Global Initiative for Chronic Obstructive Lung Disease (GOLD) the ratio of FEV_1_/FVC < 70% confirms the presence of persistent airflow limitation [30]. 

We also evaluated the FEF_25–75%_ values in this study, as low FEF_25–75%_ value might be associated with asthma [36]. The dust in the particleboard factory is made of organic particles as it comes from the Eucalyptus tree, and organic dust is known to cause asthmatic conditions [37]. However, the reason for the lack of reduced lung function variables in the particleboard factories in this study, might be that the workers had been working in the factories for very few years. The present study showed a high level of dust exposure. The geometric mean level of dust was 4.66 mg/m^3^, which is higher than the recommended limit values for inhalable wood dust of 1 mg/m^3^ [38]. Endotoxin was also documented in the particleboard factories, although the exposure levels were below the recommended health based standard of 90 EU/m^3^ set by the Dutch experts [9]. Formaldehyde was also measured in the particleboard factories, with a wide range of exposure (<0.2–5 ppm). However, the exposure time of the workers in the particleboard factories might be too short for the development of chronic lung diseases detectable by spirometry. The average service time for dust and chemicals for these workers was short, only 4 years.

To our knowledge this is the first study done among workers manufacturing particleboard from Eucalyptus to assess the prevalence of respiratory symptoms and to measure lung function status. We selected a control group from a water bottling factory, not from the general population to reduce bias that can be attributed due to baseline characteristics such as socio-demographic and economic differences. Another strength of the study is the high response rate of the participants. Furthermore, multiple regression and poisson regression was applied to control for possible confounders for lung function and respiratory symptoms. The lung function measurements were done using calibrated and sensitive portable spirometry equipment following American Thoracic Society recommendation, which should increase the validity of these results. The data collection was performed addressing one-by-one workers in a place without others listening, to reduce any possible information bias. However, there might still be a bias present, as the workers in the particleboard factory may have reported more symptoms due to their wish for an improved work environment. The size of such a bias is unknown to us. Also, the workers may have caused a recall bias in the respiratory symptom assessment, as symptoms might not be easy to remember.

Our results were based on a cross sectional study and share the limitation of this study design. The study is not conclusive concerning any cause–effect relationship between inhalable wood dust, endotoxin and formaldehyde exposure in the particleboard factories and respiratory symptoms. Longitudinal studies are needed to confirm a possible relationship between these factors. There could also be other unknown predictors present, which are not addressed in the present study [39]. Also, the study might suffer from a healthy worker effect and young age of the workers which may affect the validity of the result regarding the lung function parameters. Studies where the workers have longer service time would be of interest.

The particleboard workers that did not use proper personal protective equipment (PPE) mainly face mask during work. The absence of proper PPE in this working environment with high dust exposure may cause respiratory health problems of the workers in the future. With the present knowledge about the high dust levels in these factories, respiratory health protection is recommended among the workers, to avoid the development of any adverse health effects due to the dust exposure.

The finding is limited to male particleboard workers due to the low number of female production workers. Therefore, the finding is valid only for male particleboard workers with similar work settings in developing countries.

## 5. Conclusions

Particleboard workers in Ethiopia, exposed to wood dust, endotoxin and formaldehyde, had higher prevalence of respiratory symptoms than the controls, i.e., water bottling workers. However, lung function did not appear to be different among particleboard workers and controls. A longitudinal study is recommended to explore the cumulative impact of dust, endotoxin and formaldehyde exposure on respiratory health of particleboard workers. However, we also recommend respiratory health protection of workers with high dust exposure levels, as this type of protection was not used in the factories.

## Figures and Tables

**Table 1 ijerph-16-02158-t001:** Demographic and anthropometric characteristics of 74 particleboard workers (Exposed) and 73 water bottling workers (Controls) in Ethiopia.

Variable		Exposed	Controls	*p*-Value
Continuous variables		AM (SD)	AM (SD)	
Age (years)		28 (7)	25 (7)	0.006 ^a^
Service year (years)		4 (3)	2.2 (2)	<0.001 ^a^
Height (m)		1.70 (0.05)	1.70 (0.05)	0.634 ^a^
Weight (kg)		63 (10)	56 (6)	<0.001 ^a^
Body Mass Index		21.8 (3.1)	19.4(1.9)	<0.001 ^a^
Categorical variables		*N* (%)	*N* (%)	*p*-value
Education	Grade 1–10	17(23)	60 (82)	<0.001 ^b^
	Vocationally trained and above	57(77)	13 (18)	
Availability of separate kitchen		47 (64)	28(38)	0.002 ^b^
Uses biomass fuel for cooking		22 (30)	59 (81)	<0.001 ^b^

AM: Arithmetic Mean; SD: Standard Deviation; ^a^ Independent *t*-test; ^b^ Pearson chi square; *N*: frequency of observations (counts).

**Table 2 ijerph-16-02158-t002:** Prevalence of respiratory symptoms among 74 particleboard workers (Exposed) and 73 water bottling workers (Controls) in Ethiopia.

Variable	Exposed *N* (%)	Controls *N* (%)	*p*-Value
Cough	29 (39)	11 (15)	0.001 ^a^
Cough with sputum production	23 (31)	4 (5.5)	<0.001 ^b^
Phlegm	20 (27)	2 (2.7)	<0.001 ^b^
Wheezing	33 (45)	2(2.7)	<0.001 ^b^
Shortness of breath	18 (24)	2 (2.7)	<0.001 ^b^

*N*: frequency of observations (counts); *n*: number of study participants; ^a^ Chi square test; ^b^ Fisher exact test.

**Table 3 ijerph-16-02158-t003:** A comparison of lung function status among particleboard workers (*n* = 74) and controls (*n* = 73) in Ethiopia, both using *t*-test and multivariate regression; adjusting for age, height, previous respiratory illness, availability of separate kitchen and use of biomass fuel as source of energy.

Lung Function Parameters	Exposed	Controls	*p* ^a^	(Exposed vs. Controls) ^b^	
				β (SE)	*p*
FVC – AM (SD)	4.96 (0.37)	4.93 (0.39)	0.608	0.02 (0.03)	0.453
FEV_1_ – AM (SD)	4.10 (0.30)	4.12 (0.30)	0.743	0.012 (0.02)	0.519
(FEV_1_/FVC) × 100 – AM (SD)	82.36 (1.54)	83.14 (1.75)	0.004	−0.045 (0.11)	0.697
FEF_25–75%_ – AM (SD)	4.27 (0.37)	4.38 (0.35)	0.073	0.007 (0.02)	0.709

FVC: Forced Vital Capacity; FEV_1_: Forced Expiratory Volume in one second; FEF_25–75%_: Forced Expiratory Flow 25–75%; AM: Arithmetic Mean; SD: Standard Deviation; ^a^ Independent *t*-test; β: unstandardized Beta; SE: standard error; ^b^ multivariate analysis.

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
