# Peer review of "High Prevalence of Respiratory Symptoms among Particleboard Workers in Ethiopia: A Cross-Sectional Study"

_ijerph, 2019, doi:10.3390/ijerph16122158_

Round 1
Reviewer 1 Report
REWIEV COMMENTS (Manuscript ID: ijerph-528798)
The study explores a working situation in a specific wood industry in Ethiopia. The aim of the cross sectional work is to assess respiratory symptoms and to measure lung function in a case-control approach. The sample size used (case+control) is relatively small but sufficient for statistical purposes, and adding some information in the field of the relationships between wood dust exposure and respiratory symptoms/decreased lung function. The research design and methodology used seems appropriate and highlights an adequate bibliography. The English is clear and concise and the results may be of interest to the readers. The research topic is appropriate for IJEPH.
Some points generally minor concern needs to be addressed:
### The author stated (lines 48-52): “In previous studies from the wood industry including particleboard production, the workers have been exposed to other types of trees. ………”
Support this statement.
###Line 69-71: Include more information on the location of industries (urban context? other types of context?). Why the choice of cases in the North and the controls in the South?
### Lines 72-74 – Support
### Review table 1 (it is not clear). For example: a) evaluate presenting the descriptive statistics (Table 1) and include the inferential statistics in a second Table; b) the second part of the Table, show the term education (unclear ijn the context), while no p-value <or> 0.05 appears in the second line.
### Why was the Poisson model used instead of the logistic regression model? I think it's right in a context of counting variables, but it would be more correct to support this choice.
### Multiple linear regression was applied to analyze differences in lung function between the particleboard workers and the controls…. Have the authors tested the normality distribution (gaussian distribution) before applying the linear regression models?
### Reference: Asgedom, A.A.; Bratveit, M.; Moen, B.E. Knowledge, attitude and practice related to chemical hazards and personal protective equipment among particleboard workers in Ethiopia: A cross-sectional study. 384 BMC Public Health 2019, 19, 440, doi:10.1186/s12889-019-6807-0.
For a better understanding of the readers, the authors specify in more detail, in the manuscript, the differences between this previous research and the current work. This work appears to be carried out on the same case-control with identical n.
####In the planning of the research design, information could be included for each subject whether or not the subjects were smokers. An important control factor for statistical models.
#### Lines 277-278: “There could also be other unknown predictors present, which are not addressed in the present study. Can be supported with specific bibliography.
#### It is somewhat surprising that, despite in the manuscript (Funding) the Muhimbili University of Health and Allied Sciences (Tanzania) contributes in yhis research no one authors of this University appears in the work.
Author Response
Reviewer comment: ### The author stated (lines 48-52): “In previous studies from the wood industry including particleboard production, the workers have been exposed to other types of trees. ………” Support this statement.
Answer: We have added supporting references which used other wood types such as rubberwood, spruce, pine, oak and beech found on line 50.
The supporting references are below:
6. Thetkathuek, A.; Yingratanasuk, T.; Demers, P.A.; Thepaksorn, P.; Saowakhontha, S.; Keifer, M.C. Rubberwood dust and lung function among Thai furniture factory workers. International Journal of Occupational and Environmental Health 2010, 16, 69- 74, doi:10.1179/107735210800546281.
21. Lofstedt, H.; Hagstrom, K.; Bryngelsson, I.L.; Holmstrom, M.; Rask-Andersen, A. Respiratory symptoms and lung function in relation to wood dust and monoterpene exposure in the wood pellet industry. Uppsala Journal of Medical Sciences 2017, 122, 78- 84, doi:10.1080/03009734.2017.1285836.
Reviewer comment: ###Line 69-71: Include more information on the location of industries (urban context? other types of context?). Why the choice of cases in the North and the controls in the South?
Answer: The location of the town and inclusion of particleboard factories from both North and South is now updated as follows. “The particleboard factories are found in urban area and selected both from North and South.” Is found on line 71-72.
However, controls were selected from the northern site due to availability of water bottling factories near by the study area. It would be nice to include control group from south, but there was no nearby water bottling factory.
Reviewer comment: ### Lines 72-74 – Support
Answer: We assumed that water bottling factories had low dust level due to the nature of the activities taken place in the factory. Our assumption was verified later by the findings (see line 89-90) where particleboard workers had a geometric mean of 4.66 mg/m3 while the water bottling had 0.21 mg/m3.
Reviewer comment: ### Review table 1 (it is not clear). For example: a) evaluate presenting the descriptive statistics (Table 1) and include the inferential statistics in a second Table; b) the second part of the Table, show the term education (unclear ijn the context), while no p-value <or> 0.05 appears in the second line.
Answer: We have edited Table 1 to show clearly which variables that are continuous and categorical, respectively. A line is added to separate Education (Grade 1-10 and Vocational trained and above). The displayed p-value is for the difference in educational status among exposed and the controls.
Reviewer comment: ### Why was the Poisson model used instead of the logistic regression model? I think it's right in a context of counting variables, but it would be more correct to support this choice.
Answer: Poisson regression was used because the prevalence of the chronic respiratory symptoms (cough) was high. Prevalence ratio was chosen over the odds ratio, because the odds ratio overestimated the strength of association. Prevalence ratio can be done using Poisson regression while logistic regression is for odds ratio. The justification is stated as “Prevalence ratio (PR) was chosen instead of prevalence odds ratio (POR) due to the high respiratory symptom prevalence in this study” found on line 153-155 with a supporting reference mentioned below.
31. Tamhane, A.R.; Westfall, A.O.; Burkholder, G.A.; Cutter, G.R. Prevalence odds ratio versus prevalence ratio: Choice comes with consequences. Statistics in Medicine 2016, 35, 5730- 5735, doi:10.1002/sim.7059.
Reviewer comment: ### Multiple linear regression was applied to analyze differences in lung function between the particleboard workers and the controls…. Have the authors tested the normality distribution (gaussian distribution) before applying the linear regression models?
Answer: Yes the linearity of the data was tested. All the presented lung function parameters were normally distributed. The information is now updated as “Lung function parameters were normally distributed” found on line 147.
Reviewer comment: ### Reference: Asgedom, A.A.; Bratveit, M.; Moen, B.E. Knowledge, attitude and practice related to chemical hazards and personal protective equipment among particleboard workers in Ethiopia: A cross-sectional study. 384 BMC Public Health 2019, 19, 440, doi:10.1186/s12889-019-6807-0.
For a better understanding of the readers, the authors specify in more detail, in the manuscript, the differences between this previous research and the current work. This work appears to be carried out on the same case-control with identical n.
Answer: The previously published paper used as reference is mainly conducted to answer the research question about the workers Knowledge, attitude and practice regarding chemical hazards and personal protective equipment. The statement “….aimed to assess workers knowledge, attitude and practice regarding chemical hazards and personal protective equipment” is added which is found on line 106-108.
The submitted manuscript is aimed to assess respiratory symptoms and measure lung function among the particleboard workers and compared the finding with water bottling workers. Therefore, the sample sizes were calculated with different assumption. The published paper considered prevalence of personal protective equipment (PPE) practice reported in one of textile factory in Ethiopia (Tetemke D., A.K., Tefera Y., Sharma H.R., Worku W.,. Knowledge and practices regarding safety information among textile workers in Adwa town, Ethiopia. Science Postprint 2014, 1, 5, doi:10.14340/spp.2014.01A0004). The submitted articles sample size was calculated with different assumption as elaborated on line 97-102. The similarity of the sample size is a coincidence.
Reviewer comment: ####In the planning of the research design, information could be included for each subject whether or not the subjects were smokers. An important control factor for statistical models.
Answer: Information about smoking status was asked (see line 118) during the research. However, no smoker was recorded in both groups. The information is stated as “All of the study participants were neither smokers nor using proper personal protective devices …” found on line 183.
Reviewer comment: #### Lines 277-278: “There could also be other unknown predictors present, which are not addressed in the present study. Can be supported with specific bibliography.
Answer: Supportive reference on various risk factors for chronic respiratory diseases is reported by World Health Organization is now added on line 281-282 (reference 39).
39.World Health Organization. Risk factors for chronic respiratory diseases. Availabe online: https://www.who.int/gard/publications/Risk%20factors.pdf (accessed on 12 June).
Reviewer comment: #### It is somewhat surprising that, despite in the manuscript (Funding) the Muhimbili University of Health and Allied Sciences (Tanzania) contributes in yhis research no one authors of this University appears in the work.
Answer: The funder Norad is supporting Addis Ababa University (AAU) Ethiopia, Muhimbili University of Health and Allied Sciences (MUHAS) Tanzania and University of Bergen (UiB) Norway in a capacity building project of research and education. The project is an umbrella for different projects at AAU and MUHAS. The authors of research varies, depending on the participation in the different studies, and for the present project only researchers from AAU and UiB were involved.

Reviewer 2 Report
Interesting study attempting to find differences in pulmonary function tests in two different exposure environments. Likely reasons for finding no differences are as follow: the time of exposure may not be enough to cause measurable disease in exposed subjects who are young and healthy; we do not have longitudinal data i.e FeV1 before exposure and FeV1 after exposure. The fact that symptoms were reported is interesting.
The mean height (and standard deviation) of the experimental and control group is exactly the same in Table 1. Request authors to double check the data for accuracy. Why did the authors not used percentage predicted values for spirometry ? What data set would use for percentage predicted in the Ethiopian population.
Author Response
Reviewer comment: The mean height (and standard deviation) of the experimental and control group is exactly the same in Table 1. Request authors to double check the data for accuracy.
Answer: We have cross checked the height data set and analyzed again. However, the result remains the same.
Reviewer comment: Why did the authors not used percentage predicted values for spirometry? What data set would use for percentage predicted in the Ethiopian population.
Answer:
We did not use percentage predicted values of spirometry measurements because Ethiopia does not have a reference value for the general population.
Lung function for the general population in Ethiopia would have been useful, and it might be an aim for a future study to establish such reference values on lung function in the country.
